# It's not *what* you do, it's the *way* that you do it: An experimental task delineates among passive, reactive and interactive styles of behaviour on social networking sites

Daniel J. Shaw [1,2]*, Linda K. Kaye [3], Nicola Ngombe [4], Klaus Kessler [1,5], Charlotte R. Pennington [1,2]

1 School of Psychology, College of Health & Life Sciences, Aston University, Birmingham, United Kingdom, 2 Institute of Health & Neurodevelopment, Aston University, Birmingham, United Kingdom, 3 Department of Psychology, Edge Hill University, Ormskirk, United Kingdom, 4 Institute of Clinical Psychology and Psychotherapy, Technische Universität Dresden, Dresden, Germany, 5 School of Psychology, University College Dublin, Dublin, Ireland

* d.j.shaw@aston.ac.uk

**Data Availability Statement:** All materials and data associated with this study are publicly available on the Open Science Framework: https://osf.io/z3ac6.

## Abstract

Studies have produced vastly disparate findings when exploring relationships between social networking site (SNS) usage and psychosocial well-being. These inconsistencies might reflect a lack of consideration for *how* people use SNS; specifically, while meaningful interactions are suggested to foster positive feelings, the passive consumption of others' feeds is proposed to have negative effects on users' well-being. To facilitate the empirical evaluation of these claims, the present study developed a computerised task to measure styles of usage on a mock SNS platform. Administering this Social Network Site Behaviour Task (SNSBT) online to 526 individuals, we identified three dissociable usage styles that extend the active-passive dichotomy employed frequently in the literature: passive use (consuming content posted by others), *re*active use (reacting to others' content), and *inter*active use (interacting with others through content sharing). Furthermore, our data reveal that these usage styles differ on several measures of psychosocial variables employed frequently in the disparate literature: more interactive users reported greater feelings of social connectedness and social capital than passive or reactive users. Importantly, however, our results also reveal the multi-dimensional nature of usage styles, with online network size and time spent on SNS platforms serving as potentially confounding influences on some psychosocial measures. These findings not only advance our understanding of SNS behaviour by providing empirical support for theoretic propositions, but also demonstrate the utility of the SNSBT for experimental investigations into the psychosocial outcomes of different SNS usage styles.

**Funding:** This study was supported by an internal grant from Aston, awarded to DJS (PI), CP, KK and LK (Co-Is). The funders had no role in study design, data collection and analysis, decision to publish, or preparation of the manuscript.

**Competing interests:** The authors have declared that no competing interests exist.

# Introduction

Social networking sites (SNS; e.g., Facebook, Instagram) have become a ubiquitous feature of our daily lives; an estimated 3.6 billion people worldwide used them daily in 2020, and this is predicted to rise to 4.4 billion in 2025 [1]. By providing channels that allow people to connect socially with vast networks, many scholars consider the increased usage of SNS, and social media platforms more generally, to serve important psychosocial benefits. Nevertheless, concerns are also raised over the negative effects on our subjective well-being from SNS replacing meaningful 'real-world' interactions (see [2–4]). This debate is fuelled partly by discrepancies in research findings, which might reflect a lack of consideration for differences in the way that people engage with SNS [5]. To facilitate the reconciliation of these inconsistencies, the present study developed a computerised task to measure styles of behaviour on a mock SNS–one that can be used in future research to assess empirically the psychosocial outcomes of dissociable styles of usage. To assess the potential utility of the task for this means, we examined whether distinct usage styles differ on several psychosocial variables evaluated commonly as outcome measures in this literature: loneliness, sense of belonging, social connectedness and social capital.

Despite a wealth of research into the psychological impact of increasing SNS usage, inconsistent findings have prevented any firm conclusions being drawn. A full review of this burgeoning literature is beyond the scope of this paper, and readers are advised to consult more comprehensive overviews provided elsewhere (e.g., [2, 3, 5–8]). In brief, while a plethora of studies report that social media usage is associated with increased social connectedness and reduced loneliness [9–12], others have observed detriments to loneliness and well-being from greater use of such platforms (e.g., [13–15]) or report no meaningful relationships between social media use and indices of psychological well-being (e.g., [16–19]).

According to the interpersonal-connection-behaviours framework [20], such disparities likely reflect differences in styles of social media usage: while directed communication, posting, and sharing of content (active usage) are believed to enhance users' psychological well-being by increasing social connectedness and decreasing loneliness [21], non-communicative consumption of feeds (passive usage) is said to increase experiences of isolation and decrease self-esteem (e.g., [22, 23]; for a review see [24]). This highlights the need for accurate measurements of user behaviour on SNS if we are to understand the psychosocial outcomes associated with increasing usage ([8, 25]; see also [26]).

A recent meta-analysis has revealed the vast number of self-report instruments that have been developed for measuring styles of SNS usage along the active-passive dichotomy [25]. Recently, however, many scholars have questioned the usefulness of this crude categorisation, calling for a more nuanced refinement (e.g., [5]). It is also argued that the field of cyberpsychology needs to move beyond subjective self-report questionnaires and consider behavioural methods that allow for more direct assessments of usage styles on SNS [18]. One alternative method is to use digital tracking with data recorded directly from devices [27–30]. Importantly, though, while such direct tracking of activity is considered widely to be the gold standard method (see [17, 31]), the necessary reliance upon the willingness of individuals to allow researchers to monitor their (natural) usage raises questions concerning sampling bias [32]. Further, tracking measurements of SNS usage will vary markedly across software platforms and measurement periods (e.g., weekday *vs.* weekend) and become inaccurate when individuals have access to such sites through multiple unlinked devices [32].

Assessing social media usage objectively whilst avoiding these limitations is achievable with a behavioural measure that can be administered under controlled experimental conditions. To capture different styles of usage that might relate differentially to user outcomes, such a

measure must go beyond unidimensional assessments of engagement (e.g., number of log-ins, activity duration) and delineate among individuals who exhibit different behavioural patterns [25]. As reviewed above, evidence suggests that active engagement with others on SNS is more likely to be associated with positive psychosocial measures when compared with passive use. More recent research suggests that usage styles can be divided further, however; a recent meta-analysis distinguished between one-sided (e.g., liking) and bidirectional active social exchanges (commenting) in computer-mediated communication [33]. Similarly, other research has delineated between public (e.g., bidirectional chatting) and private (e.g., tagging) active engagement [8]. If the degree to which individuals engage in meaningful *inter*actions with other users is indeed a key factor in the positive outcomes of active SNS usage, as proposed by the interpersonal-connection-behaviours framework [20], a useful behavioural measure should be capable of dissociating empirically among these passive, reactive (unidirectional) and interactive (bidirectional) forms of usage.

To investigate if these more nuanced styles of usage can be identified behaviourally, the present study developed the Social Networking Site Behaviour Task (SNSBT)–a mock SNS platform based loosely on Facebook and Instagram. As a means of validating this task, we then examined if dissociable patterns of reactive and interactive behaviour converged with subjective perceptions of usage style assessed with a commonly used self-report instrument based on the active-passive dichotomy [34], which we modified to distinguish further between reactive and interactive usage. Finally, we examined whether distinct usage styles identified on this mock SNS differed in terms of several psychosocial factors employed frequently in the disparate literature. Importantly, this was not to assess the outcomes of dissociable styles, which requires pre- and post-usage comparisons or real-time measurements (e.g., [13]); rather, this evaluated the potential utility of the SNSBT for future research and reconciling inconsistent findings. First we assessed differences in self-reported *loneliness* and *sense of belonging*, the latter referring to the degree to which an individual values their involvement in, or feels valued by others within a social system [35]. Since social interactions are important for satisfying socioemotional needs, including the need for belonging (see [20]), we predicted that greater interactive usage on the SNSBT would be associated with less loneliness and an increased sense of belonging relative to more reactive and passive usage. Similarly, we investigated relationships between patterns of usage on the mock SNS and self-reported *social connectedness*–the experience of belonging to a social network or community [36], and *social capital*–the perceived outcomes (e.g., emotional support) that individuals take from online relationships with close friends and family (bonding) or more extended networks (bridging; [37]). Given the emerging evidence for opposing associations between self-reported active or passive usage and these psychosocial outcomes (see [24, 33]), we predicted that subjective reports of social connectedness and social capital would be greater from individuals showing more interactive usage on the SNSBT compared with reactive or passive usage. Finally, as an exploratory analysis, we investigated whether dissociable styles of behaviour on our experimental task differed on other subjective measures of SNS usage shown previously to relate to psychosocial variables —namely, the size of one's online social network [38, 39] and the number of hours spent on SNS platforms [40].

## Method

### Participants

Participants were recruited through Prolific Academic (www.prolific.co), which has been shown to avoid some of the shortcomings associated with other crowdsourcing platforms [41]. To be eligible for participation, individuals had to be (1) over the age of 18, (2) a Facebook

user, and (3) able to complete the study on a PC or laptop with a standard QWERTY keyboard in a single continuous session. Participation was recompensed at £7.50/hour. In total, 622 Prolific identifiers were initially recruited, a sample size determined solely by available funding. Of these, 25 individuals were removed because they had participated more than once. Of the remaining 596, we removed 69 who provided incomplete data across measures (see below) and two who reported no prior Facebook use. The final sample therefore comprised 526 participants ($M_{AGE}$ = 31.07 years, $SD$ = 11.41, range = 18–71; 254 males). Sensitivity power analyses were conducted using the *pwr* package in R-Studio [42]; based on the cluster analyses described below, these indicated that we were able to detect effects of Cohen's $d > .28$ with 80% power at $\alpha$ = .05.

All volunteers provided written informed consent and the study was approved by the University Research Ethics Committee at Aston University (ref. 1649).

## Procedure

Participants completed a large battery of measures comprising the Social Networking Site Behaviour Task and five self-report instruments. The former was administered through Pavlovia (https://pavlovia.org)–an online platform for running PsychoPy experiments (v3; [43]). Upon completion of this task, participants were re-directed automatically to Qualtrics (www.qualtrics.com) where they completed the five questionnaires. The SNSBT task was always completed first to avoid any priming from explicit questionnaire-based content, and the subsequent self-report instruments were presented randomly to minimise order effects. The entire experiment lasted approximately 45 minutes. To ensure data integrity, an attention check was embedded among the questionnaires ("How many months are there in a year?") that required participants to select one of four possible answers. All participants answered this check correctly.

## Measures

**Social Networking Site Behaviour Task (SNSBT).**   We developed the SNSBT to assess whether passive, reactive, and interactive behavioural tendencies could be identified on a mock SNS. Before the task, participants were informed that they would be connected to a network of 99 other 'friends' on a new social networking platform and would see a series of images posted by other members of that network. In response to each image, they were asked to either "Like" or "Share" it, or skip to the "Next" image. Participants were also told that, since not all members of this network were 'friends' with every other member, the images they chose to share would be seen for the first time by some people who might then comment on the shared post. Finally, to ensure their behaviour was representative of their typical SNS behaviour, participants received the following instruction:

> "*Please take a moment to think about how you typically respond on social media (e.g., Facebook). Do you "like" a lot of pictures, do you often "share" them, or do you usually scroll through images without liking or sharing them? We want you to act in the same way on this task as you would do on other social media platforms*".

The images comprised 90 pictures of natural landscapes selected from the Nencki Affective Picture System (NAPS; [44, 45])–a validated and standardised set of high-quality photographs. Non-social images were selected purposefully to ensure that (un)familiarity of agents within the scenes could not influence responses. We used normative ratings provided for the NAPS to ensure that selected images were of neutral-positive valence ($M$ = 7.12, $SD$ = 0.74;

# Trial structure

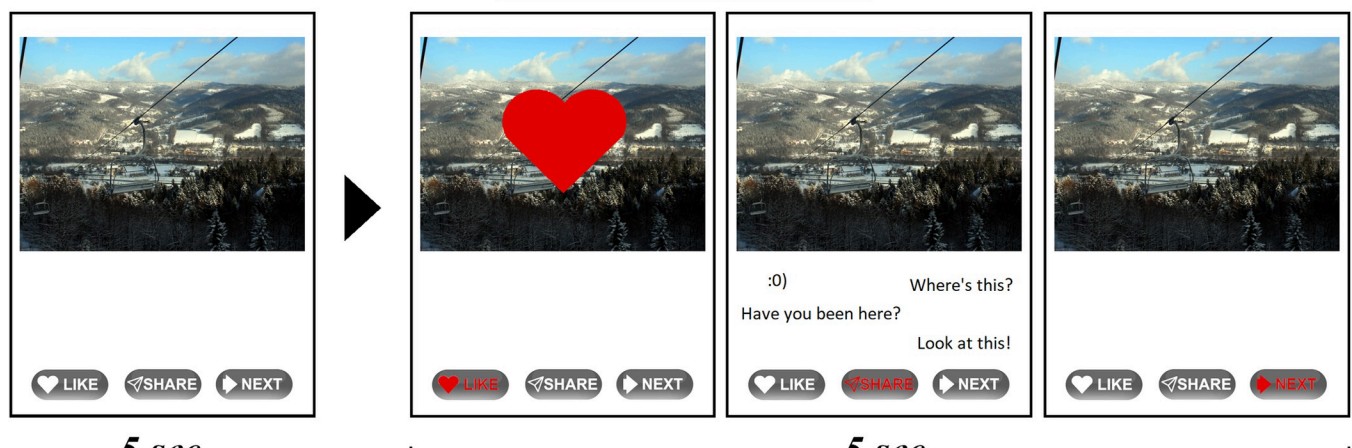

# Example images

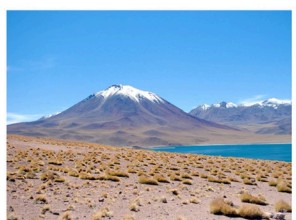 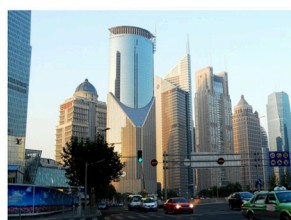 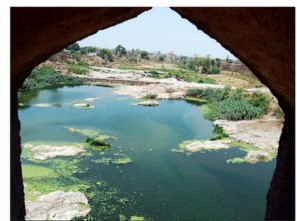 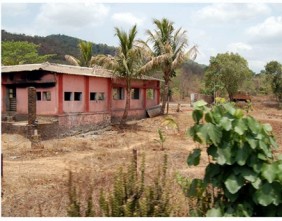

**Fig 1. The Social Networking Site Behaviour Task (SNSBT).** The top row illustrates the trial structure and three possible response options. The bottom row provides example images selected from the Nencki Affective Picture System. The example images are reprinted from the Nencki Affective Picture System [44] under a CC BY licence, with permission from Artur Marchewka, original copyright 2014.

range = 5.09–8.54) and low arousal ($M$ = 3.59, $SD$ = 0.67; range = 2.04–4.90). These normative ratings did not differ significantly between females and males ($p$ > .05).

Images were presented in a random sequence for a total of 90 trials. Participants were given 5 seconds to view and respond to each image by choosing "Like", "Share" or "Next" (key 1, 2, or 3 on a standard computer, respectively). Upon a response, their selection was highlighted for a subsequent 5 seconds before the subsequent image appeared. When participants chose to "Like" an image, a symbol of a heart was presented to indicate that their reaction had been posted to the network. After choosing to "Share" an image, participants received immediate feedback in the form of three comments presented in short succession that were provided ostensibly by other members of the network. These comments were selected from publicly available Facebook content and matched meaningfully to each image (e.g., "*Have you been here?*", "*I want to take my family here*", "*I love this place!*"). When participants chose "Next", the next image was displayed. Fig 1 depicts the trial structure. Responses to the images were shown to have excellent internal reliability (*a* = .96). For each participant, we calculated the proportion of "Like", "Share" and "Next" responses across all 90 trials.

**Self-reported social media usage.** As a subjective measure of behaviour on social networking platforms, we adapted an existing self-report instrument that has been shown previously to dissociate between active and passive social media use (SMU; [34]). Importantly, the

original seven-item version of this instrument does not differentiate between *re-* and *inter-*active styles; the four items that ask about liking, commenting, and sharing tendencies are sub-sumed under "active" usage. To assess whether these more nuanced styles of active SMU can be detected in subjective reports, we added two items such that passive, reactive, and interac-tive styles were each assessed with three distinct items (see Qualtrics Questionnaires; https://osf.io/z3ac6). Further, to avoid a potential ceiling effect from the temporal judgments used previously as response options (e.g., "*Several times a day*"), we used labels of frequency on a 5-point Likert scale (1 = *Never*, 5 = *Very Often*).

To assess the factor structure of this modified instrument, we performed Principal Compo-nent Analysis (PCA) with varimax rotation. Contrary to our expectations of a three-factor structure, this revealed an optimal two-factor solution (determinant of correlation matrix = .05; Kaiser-Meyer-Olkin = .81) replicating the original findings [34]. Six items measuring reac-tive and interactive behaviours accounted for 31.66% of variance and the remaining three assessing passive usage explained 25.66% (see S1 File). In light of these factor loadings, we label these as "Active" and "Passive" sub-scales. Both sub-scales demonstrated acceptable internal reliability ($\alpha$ = .80 and .75, respectively). Next, we computed a single index that reflected an individual's self-reported tendency for active relative to passive SMU. Specifically, we calcu-lated the difference between their mean rating across all items loading onto the Active factor and those loading onto the Passive factor; greater scores on this index of usage (SMU$_{diff}$) indi-cate more active relative to passive usage. This relative measure was chosen to account for prior research findings; namely, studies have shown that while all users of social media engage most frequently in passive behaviours, they differ in their *relative* active engagement [46]. Indeed, the mean value of this difference measure across the sample was -.94 (SD = .83; range -3.82–1.67) indicating that participants reported more passive than active SNS usage.

**Loneliness.**   Loneliness was measured using the eight-item UCLA Loneliness Scale-Short Form [47]. Participants respond to each item (e.g., "There is no one I can turn to") on a 4-point Likert scale anchored between 1 (*Never*) and 4 (*Always*). This measure had good inter-nal consistency ($\alpha$ = .87) and a total score across all items was computed, with higher scores representing greater feelings of loneliness. The mean across the sample was 17.09 (SD = 5.26; range 8–32).

**Sense of belonging.**   Social belonging was measured using the Sense of Belonging Instru-ment (SOBI-P; [35]). This 18-item scale contains questions such as "I often wonder if there is any place on earth where I really fit in", each of which is responded to on a 4-point Likert scale ranging between 1 (*Strongly Disagree*) and 4 (*Strongly Agree*). All items on the SOBI-P ques-tionnaire are framed negatively, with higher scores representing lower social belonging. To aid interpretation alongside our other measures, we reverse scored the questionnaire so that higher scores represent greater sense of belonging. Responses were found to have excellent internal consistency ($\alpha$ = .95) and a total score was computed across items with higher scores repsenting greater social belonging. The mean across the present sample was 54.06 (SD = 13.21; range 19–72).

**Social connectedness.**   Social connectedness was measured using the Social Connected-ness Scale [36]. This questionnaire comprises 20 items (e.g., "I feel close to people"), and responses are provided via a 6-point Likert scale ranging from 1 (*Strongly Disagree*) to 6 (*Strongly Agree*). This measure resulted in excellent internal reliability ($\alpha$ = .95) and a total score was computed across items, with higher scores representing greater feelings of social connectedness. The mean score for the present sample was 80.75 (SD = 19.59; range 23–120).

**Online social capital.**   Online social capital was measured using the Internet Social Capital Scale [37]. The original instrument comprises 40 items that ask individuals about both their online and offline social bonding and bridging experiences, but we focus exclusively on the

online subscales that together comprise 20 items (10 items for bonding, 10 for bridging). For the current study, these subscale items were adapted to be conceptually correspondent with Facebook use (e.g., "When I feel lonely, there are several people on Facebook I can talk to"). Responses were made on a 5-point Likert scale ranging from 1 (*Not characteristic of me*) to 5 (*Extremely characteristic of me*). Both the bonding ($\alpha$ = .90) and bridging ($\alpha$ = .91) sub-scale resulted in excellent internal reliability. A total score was computed across items representing each subscale, with higher scores representing greater online social capital. Across the sample, the mean scores for the bonding and bridging subscale were 29.91 (*SD* = 9.44; range 10–50) and 31.10 (*SD* = 9.35; range 8–50), respectively.

### Exploratory measures

**Online social network size.**   In addition to the aforementioned confirmatory measures which allowed us to evaluate our hypotheses directly, the online protocol also included three questions employed in another study [38] to assess whether styles of behaviour on the SNSBT were associated with the size of participants' online social networks. Participants first responded to the questions "How many Facebook friends do you have?" and "How many friends do you have in your offline social circle?". They then responded to the question "What percentage of Facebook friends do you consider to be genuine friends?", but we do not consider answers to this question in the analyses that follow. Higher scores represent greater social networking size. The average number of Facebook friends was 415.38 (*SD* = 448.58), of whom 27.09% (*SD* = 25.05) were considered as genuine friends. Participants also reported an average of 27.40 (*SD* = 59.07) offline friends. We used participants' total number of online friends (*Friends*) as an exploratory covariate in our analyses.

**Facebook hours.**   To assess whether styles on usage on the SNSBT were also associated with the number of hours spent on SNS platforms, we asked participants two questions that measured their Facebook usage on a typical weekday ("On average, how much do you use Facebook on a weekday [Monday-Friday]?") and weekend ("On average, how much do you use Facebook on a weekend day [Saturday & Sunday]?"). Given that self-reported usage was much greater during the weekend than during the weekdays and provided greater variability, we chose to focus on this particular measure of usage frequency (*Hours*) as an exploratory covariate.

### Analytic strategy

To assess whether dissociable styles of behaviour could be detected on the SNSBT, we first conducted a two-step cluster analysis on participants' response patterns; namely, the proportion of trials in which they made a "Like", "Share" or "Next" response. This particular clustering technique is considered more objective than alternatives because it determines cluster membership based upon statistical measures of fit (e.g., Bayesian Information Criterion [BIC]) and is capable of handling large datasets [48]. In line with published recommendations [49], we used the BIC index to determine the model fit and the log-likelihood to assess the probability that each observation (participant) belongs to a cluster.

Having identified the optimal number of clusters, next we performed a series of ANOVAs to ascertain the response tendencies of constituent members on the SNSBT and compared this to their self-reported usage (SMU$_{diff}$). We then examined whether dissociable styles of behaviour on the task emerging from the cluster analysis differed on any of the psychosocial measures. In follow-up exploratory analyses, we examined whether types of usage on the SNSBT differed based on the number of friends and the number of hours spent on Facebook reported by individuals comprising each cluster. Given that the former was recorded in an ordinal

fashion, this was assessed with non-parametric tests. Where the clusters differed on *Friends* and *Hours*, they were entered as (dummy-coded) covariates in follow-up ANCOVA analyses to determine if discrete patterns of SNS usage differed on any of the psychosocial measures independent of these covariates.

## Results

All analyses were performed in SPSS (v.26). In the following sections, means are presented with standard deviations (*SD*) and the results of all pairwise comparisons are reported after Bonferroni correction ($p_{corr}$). Effect sizes are reported as partial eta squared for main effects, and Cohen's *d* for mean differences [50].

### Cluster analysis

The BIC change ratio indicated an optimal three-cluster solution. A Silhouette measure of cohesion and separation indicated good model fit (.60), and the ratio of distance measures was acceptable (2.09). Each cluster differed significantly in terms of the proportion of "Next" (*F* [2,523] = 1403.67, $p < .001$, $\eta_p^2 = .84$), "Like" (*F*[2,523] = 505.69, $p < .001$, $\eta_p^2 = .66$) and "Share" (*F*[2,523] = 246.09, $p < .001$, $\eta_p^2 = .49$) responses: Participants in Cluster 1 made the highest proportion of "Next" but the fewest "Like" and "Share" responses ($p_{corr} < .001$), while those in Cluster 3 made the least number of "Next" but the highest proportion of "Like" and "Share" responses ($p_{corr} < .001$). Those in Cluster 2 made an intermediate proportion of "Next", "Share" and "Like" responses relative to the other two clusters ($p_{corr} < .001$). Based on these response patterns, we refer to Cluster 1 herein as *Passive*, Cluster 2 as *Reactive*, and Cluster 3 as *Interactive*. The response patterns of each cluster are illustrated in Fig 2.

### Cluster comparisons

A one-way ANOVA comparing the three clusters on the SMU$_{diff}$ variable revealed that, while all participants considered themselves to exhibit stronger passive than active usage on Facebook (indicated by negative scores on this relative measure), these self-perceptions differed significantly between each cluster in line with their behaviour on the task (*F*[2,523] = 14.42, $p < .001$, $\eta_p^2 = .05$). Individuals in the *Passive* cluster produced the lowest score (-1.14 [.85]), indicating the strongest self-perceived passive tendency, while the *Interactive* cluster scored the highest (-.65 [.77]) and members of the *Reactive* cluster scored at an intermediate level (-.93 [.80], $p_{corr} < .05$).

   When comparing the clusters on each of the psychosocial measures, a monotonically decreasing pattern emerged for the *Bridging* subscale of *Social Capital* (*F*[2,523] = 17.80, $p < .001$, $\eta_p^2 = .06$); the *Interactive* cluster scoring significantly higher (34.85 [8.35]) than the *Reactive* (30.82 [9.06]; $p < .001$, $d = .46$) and *Passive* clusters (28.88 [9.51]; $p < .001$, $d = .67$), and individuals in the *Reactive* cluster scoring significantly higher than their *Passive* counterparts ($p = .035$, $d = .21$). There was also a main effect of Cluster on the *Bonding* subscale (*F*[2,523] = 3.53, $p = .030$, $\eta_p^2 = .01$), whereby individuals in the *Interactive* cluster scored higher (31.61 [9.61]) than those in the *Passive* cluster (28.84 [9.77], $p = .025$, $d = .29$), but there were no differences between those in the *Reactive* cluster (29.85 [8.80]) and either the *Interactive* or *Passive* clusters ($p > .296$, $d = .19$ and .11, respectively). A similar pattern emerged for self-reported *Social Connectedness* (*F*[2,523] = 3.38, $p = .035$, $\eta_p^2 = .01$); those in the *Interactive* cluster scored significantly higher (84.29 [20.77]) than the *Passive* cluster (78.70 [20.13]; $p = .030$, $d = .27$), but there were no significant differences between the *Interactive* and *Reactive* (80.45 [17.77]; $p = .246$, $d = .20$) or the *Reactive* and *Passive* clusters ($p > .999$, $d = .09$). There was no significant difference between the clusters in terms of *Sense of Belonging* (*F*[2,523] = .153, $p = .858$, $\eta_p^2 = .001$) or *Loneliness* (*F*[2,523] = .174, $p = .840$, $\eta_p^2 = .001$).

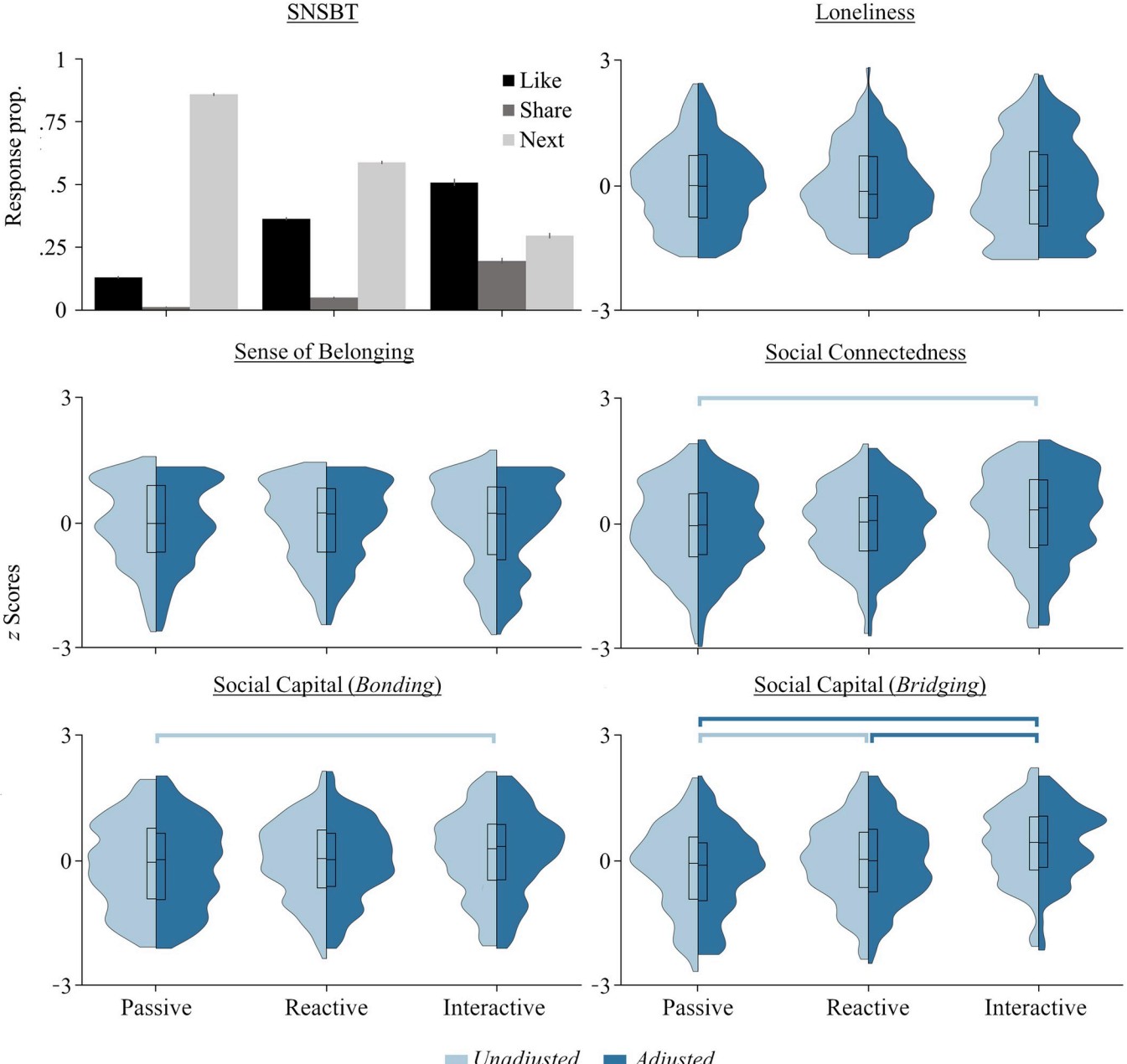

**Fig 2. Behaviour on the experimental task and self-report ratings across the psychosocial variables for each cluster.** Clockwise from top-left: Mean (±SE) proportions of the three possible response options on the SNSBT across individuals comprising each cluster; distributions of ratings within each cluster for the UCLA Loneliness Scale, Sense of Belonging Instrument, Social Connectedness Scale, and Internet Social Capital Scale. *Note*: Split violin plots present z-scored raw (unadjusted) and residualised ratings from the ANCOVA (adjusted for both the *Friends* and *Time* covariates). Plots are cut at minimum and maximum values, with inner boxplots presenting medians, first and third quartiles. C1 = cluster #1 (*Passive*), C2 = cluster #2 (*Reactive*), C3 = cluster #3 (*interactive*). Horizontal lines indicate significant differences between clusters for raw and adjusted ratings (light or dark blue, respectively).

An exploratory ANOVA revealed that the three clusters also differed significantly in the number of self-reported Facebook friends ($F[2,523] = 4.96$, $p = .007$, $\eta_p^2 = .02$): Individuals comprising the *Interactive* cluster reported significantly more *Friends* (486.65 [526.85]) than those the *Passive* cluster (341.58 [345.43], $p_{corr} = .010$, $d = .33$), while members of the *Reactive* cluster (444.98 [477.28]) did not differ significantly from their *Interactive* ($p_{corr}>.999$, $d = .08$)

**Table 1. Cluster characteristics across all measures.**

|  | Cluster 1 (*Passive*) | Cluster 2 (*Reactive*) | Cluster 3 (*Interactive*) |
|---|---|---|---|
| N (%) | 205 (39.0) | 186 (35.4) | 135 (25.7) |
| Males:Females | 100:105 | 79:102 | 75:59 |
|  | Means (±*SD*) | | |
| Age | 31.30 (11.20) | 30.65 (10.94) | 31.32 (12.39) |
| Next | .86 (.08) | .59 (.08) | .30 (.13) |
| Like | .13 (.07) | .36 (.09) | .51 (.17) |
| Share | .01 (.02) | .05 (.05) | .20 (.14) |
| SMU$_{diff}$ | -1.14 (.85) | -.94 (.80) | -.65 (.77) |
| Bridging* | 29.18 (8.96) | 30.84 (8.93) | 34.44 (8.96) |
| Bonding* | 29.04 (9.42) | 29.80 (9.37) | 31.31 (9.42) |
| Belonging* | 53.83 (13.24) | 54.52 (13.18) | 54.13 (13.24) |
| Social Connectedness* | 79.03 (19.49) | 80.38 (19.38) | 84.16 (19.47) |
| Loneliness* | 17.22 (5.30) | 17.01 (5.26) | 16.89 (5.28) |
| Friends | 341.58 (345.43) | 444.98 (477.28) | 486.65 (526.85) |
| Hours** | 1–2 hours | 1–2 hours | 2–3 hours |

Note:

*Means are adjusted for covariates (see text for details).

**Median self-reported hours spent on Facebook on a typical weekend.

or *Passive* counterparts ($p_{corr}$ = .067, *d* = .25). Furthermore, a non-parametric Kruskall-Wallis test demonstrated that the clusters differed significantly in the ordinal number of weekend hours they spent on Facebook (H[2] = 35.01, *p* < .001, $\varepsilon^2$ = .06]). Follow-up Mann-Whitney tests showed that the *Interactive* cluster reported spending more *Hours* on Facebook at the weekend (median = 2–3 hours) than the *Reactive* (median = 1–2 hours; *Z* = 2.56, *p* < .001) and *Passive* clusters (median = 1–2 hours; *Z* = 5.69, *p* < .001), and the *Reactive* cluster reported more *Hours* than the *Passive* users (*Z* = 3.70, *p* < .001). Given that the clusters differed in terms of both the *Friends* and *Hours*, these were added as covariates in exploratory ANCOVA analyses to assess their potentially confounding influence on the aforementioned differences between usage styles and psychosocial measures. The adjusted means estimated from this ANCOVA, together with other cluster characteristics, are presented in Table 1.

When re-evaluating the *Bridging* subscale of *Social Capital*, although both *Friends* and *Hours* were identified as significant covariates (*F*[1,516] = 15.21, *p* < .001, $\eta_p^2$ = .03; *F*[1,516] = 6.03, *p* = .014, $\eta_p^2$ = .01, respectively), the main effect of *Cluster* remained significant (*F*[2,516] = 13.81, *p* < .001, $\eta_p^2$ = .05). Here, a similar (though not identical) pattern of differences emerged: while the *Interactive* cluster still scored significantly higher (34.44 [8.96]) than both the *Reactive* (30.84 [8.93]; *p* = .001, *d* = .40) and *Passive* clusters (29.18 [8.96]; *p* < .001, *d* = .59), there was no longer any significant difference between the *Reactive* and *Passive* clusters (*p* = .206, *d* = .19). For the Bonding subscale of *Social Capital*, *Friends* again served as a significant covariate (*F*[1,516] = 5.27, *p* = .022, $\eta_p^2$ = .01) while *Hours* did not (*F*[1,516] = 3.71, *p* = .055, $\eta_p^2$ < .01). More importantly, the addition of these factors rendered the main effect of *Cluster* non-significant (*F*[2,516] = 2.31, *p* = .100, $\eta_p^2$ < .01). Similarly, *Friends* was again a significant covariate when re-evaluating the influence of cluster membership on *Social Connectedness* (*F*[1,516] = 8.78, *p* = .003, $\eta_p^2$ = .02) while *Hours* was not (*F*[1,516] = 2.04, *p* = .154, $\eta_p^2$ < .01), and the main effect of *Cluster* no longer remained significant with the addition of these covariates (*F*[2,516] = 2.83, *p* = .060, $\eta_p^2$ = .01). For *Sense of Belonging*, *Hours* was identified as a significant covariate (*F*[1,516] = 5.24, *p* = .023, $\eta_p^2$ = .01) but *Friends* was not (*F*[1,516] = 2.25,

$p = .134$, $\eta_p^2 < .01$), and the main effect of *Cluster* remained non-significant ($F[2,516] = .13$, $p = .876$, $\eta_p^2 = .001$). For *Loneliness*, neither *Friends* ($F[1,516] = .1.53$, $p = .217$, $\eta_p^2 < .01$) nor *Hours* were identified as significant covariates ($F[2,516] = .17$, $p = .241$, $\eta_p^2 < .01$) and the non-significant effect of *Cluster* remained ($F[2,516] = .17$, $p = .842$, $\eta_p^2 = .001$). Refer to Fig 2 for response patterns for each cluster after the addition of the *Friends* and *Hours* covariates.

## Discussion

The present study set out to develop a computerised task capable of delineating among distinct styles of usage on a mock social networking site (SNS), and evaluated its utility for future research into the psychosocial outcomes of different usage styles that might reconcile some of the discrepant findings in this field. Specifically, we designed the Social Networking Site Behaviour Task (SNSBT) as a tool with which researchers can evaluate the claim that interactive SNS usage (content sharing) will lead to more positive psychosocial outcomes than reactive (content liking) and passive usage (content scrolling). Data acquired online from this sample revealed that behaviour on the SNSBT can indeed be classified reliably into these three distinct styles of usage. Comparisons among these styles revealed that individuals exhibiting the strongest tendency for interactive usage reported stronger feelings of social connectedness and scored higher on both sub-dimensions of social capital relative to those expressing more reactive or passive styles. Importantly, however, those who showed the greatest amount of interactive use on the SNSBT also reported the greatest number of Facebook friends and the most amount of time spent on SNS compared with the other two styles. When comparisons among usage styles were corrected for these factors, only the difference between the bridging sub-dimension of social capital remained. We now discuss the implications of these findings for future research in the field of cyberpsychology.

Disparate findings emerging from research into the psychosocial outcomes of SNS usage are attributed frequently to an active-passive dichotomy of behaviour (see reviews by [5, 24]). More recently, however, scholars have questioned the usefulness of such a crude binary distinction, arguing instead for more nuanced classifications that capture with greater accuracy the multi-dimensional nature of behaviour on SNS [8]. For example, unlike the broad definition of active use that includes not only commenting and sending messages but also content liking [24], other scholars differentiate between two-way "truly" interactive (e.g., a continuous message exchange) and two-way reactive usage (e.g., content liking; [33]). Similarly, the interpersonal-connection-behaviours framework [20] makes a distinction between connection- and non-connection-promoting active behaviours on SNS; the former involves interactions whereby users show responsiveness and care towards one another's needs, while the latter encompasses active usage that fails to contribute to interpersonal connection. Behaviour on the SNSBT provides the first empirical data in support of these finer differentiations: Individuals were dissociated according to their tendency for passive, *re*active, or *inter*active usage, the last of these expressed through content sharing that exposed users to bidirectional exchanges through the feedback they received. These data-driven classifications of usage on the SNSBT also aligned with users' self-reported tendencies on SNS: whilst all participants perceived their usage of social media to be primarily passive, aligning with previous findings [51], subjective estimates of active usage were greater in individuals with higher proportions of Share responses compared with Like and Next responses. However, responses of the SNSBT were more internally reliable than individuals' subjective reports. As such, this computerised task provides researchers with a reliable proxy measure of behaviour on SNS that may overcome the limitations of self-report data [5, 25].

Individuals exhibiting more interactive usage reported stronger feelings of connectedness and social capital compared with those who showed more reactive or passive behaviour, and

those with a more reactive style of usage reported feeling more connected than those expressing a tendency for passive usage. It is important to stress that these results of comparisons between usage styles on subjective ratings of psychosocial variables provides only a measure of the potential utility of this novel task for future research; they provide no indication of the *outcomes* of SNSBT usage styles. To measure the outcomes of different usage styles on this task, future studies should employ the experimental designs used elsewhere; for example, they might measure changes in users' mood [52, 53] or subjective well-being [22, 54] before and after specific types of engagement on this task. Alternatively, to capture the outcomes of spontaneous usage styles, future research might employ real-time ecological measurements of psychosocial variables *during* engagement with the SNSBT (see [13].

Perhaps more importantly, our results also revealed that while psychosocial variables differ among individuals according to their response tendencies on the SNSBT, this was often accounted for by the number of friends that participants reported having on Facebook and the amount of time they spent on the platform. This converges with other studies showing that the number of friends and time spent on SNS are associated strongly with social satisfaction and well-being (e.g., [10, 39, 40]; but see [55]). In this light, our findings indicate that usage style operates in concert with, or even emerges as a function of, online network size and time spent on SNS. Indeed, psychological theories of friendship (e.g., [56, 57]) posit that while social networks provide an opportunity for emotional closeness and social support, the degree to which an individual experiences this is dependent upon the intensity and intimacy of relationships they maintain with their network. Larger online social networks reported by interactive users will likely enable them to have more meaningful (bidirectional) exchanges with a wider network of close and casual friends, allowing them to exercise their preferred style more frequently and benefit from it maximally.

As with all experimental measures, the experimental control afforded by the SNSBT as implemented in the current study offers only a limited insight into the complex and multidimensional range of behaviours shown on SNS. However, this limitation opens up exciting avenues of research for the field of cyberpsychology. First, like the majority of earlier studies that have investigated differential associations between active and passive usage and psychosocial factors, our experimental task resembles a hybrid of the Facebook and Instagram platforms. As such, it (currently) measures users' engagement with public image-based content only, not private text-based interactions. Since image-based platforms (e.g., Instagram) appear to have different influences on our well-being compared with text-based alternatives [58], future studies should consider adapting the SNSBT to capture usage styles across alternative platforms. Second, participants in our study received only positive feedback after sharing content with their network. Future work might manipulate this to evaluate the notion that increased well-being from (inter-)active SNS usage is underpinned by positive affective experiences brought about by commenting and chatting [59]. In other words, does negative feedback from posting and sharing content serve to undermine the potential benefits of interactive usage? Similarly, participants in the present study viewed and responded only to positively valenced images of non-social landscapes posted ostensibly by anonymous members of an online network. Although these procedural decisions were made intentionally to avoid multiple confounding factors on participants' behaviour, the SNSBT presents a tool with which to investigate the impact of more social material on styles of usage. For example, future studies might use different categories of stimuli to build upon research that has explored interactions between usage styles and type of content (e.g., [60]).

Finally, although the current instantiation of the SNSBT incorporated some characteristics of real SNS platforms (i.e. content scrolling, liking, sharing), it did not expose users to other important design features that have been shown to cue habitual social media usage patterns.

For example, the hybrid layout of the mock platform differed from either Facebook or Instagram, thereby removing contextual cues that might activate stimulus-response associations that drive particular usage styles (e.g., within-platform chat sounds). Similarly, the non-emotional images used in the current version of the task differ considerably from the unexpected and often emotional content through which users scroll on real SNS platforms, which are considered to be more rewarding and capable of triggering habitual behaviours acquired through reinforcement learning [61–63]. As such, the dissociable styles of usage that we have identified with the SNSBT may not perfectly reflect users' real-world habits on SNS platforms. However, the ease with which the layout, interface and content of the SNSBT can be modified presents a tool with which these potential drivers of social media habit formation can be investigated experimentally.

## Conclusion

We have developed a simple, publicly available computerised task that provides a new method with which to objectively quantify different styles of behaviour on social networking sites (SNS). The versatility of this task enhances its utility for future research in the field of cyberpsychology; it can be adapted easily to measure usage styles across multiple SNS platforms, to assess content and design features of SNS that might influence discrete usage styles, and perform experimental investigations into the influence of different SNS usage styles on psychosocial outcomes.

## Supporting information

**S1 File. Exploratory factor analysis of the "passive and active social media use" questionnaire.** This presents the inter-item correlation matrix, factor loadings and component plot, which reveal that a two-factor solution is optimal for responses to the Social Media Usage questionnaire [34].
(DOCX)

## Author Contributions

**Conceptualization:** Daniel J. Shaw, Linda K. Kaye, Charlotte R. Pennington.

**Data curation:** Daniel J. Shaw, Charlotte R. Pennington.

**Formal analysis:** Daniel J. Shaw, Charlotte R. Pennington.

**Funding acquisition:** Daniel J. Shaw, Linda K. Kaye, Klaus Kessler, Charlotte R. Pennington.

**Methodology:** Daniel J. Shaw, Charlotte R. Pennington.

**Project administration:** Daniel J. Shaw.

**Resources:** Daniel J. Shaw.

**Software:** Daniel J. Shaw.

**Supervision:** Daniel J. Shaw.

**Validation:** Daniel J. Shaw.

**Visualization:** Daniel J. Shaw, Nicola Ngombe.

**Writing – original draft:** Daniel J. Shaw.

**Writing – review & editing:** Linda K. Kaye, Nicola Ngombe, Klaus Kessler, Charlotte R. Pennington.

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
