## [Decision Letter · Decision Letter 0]

25 Aug 2022

PONE-D-22-09916It’s not what you do, it’s the way that you do it: An experimental task delineates among styles of behaviour on social networking sites and psychosocial measures.PLOS ONE

Dear Dr. Shaw,

Thank you for submitting your manuscript to PLOS ONE. After careful consideration, we feel that it has merit but does not fully meet PLOS ONE’s publication criteria as it currently stands. Therefore, we invite you to submit a revised version of the manuscript that addresses the points raised during the review process.

 All the reviewers as well as myself see promise in your manuscript. I do think some minor revision is warranted, and please pay particular attention to the requests of the 2nd reviewer to tone down the expectations of wellbeing as an associated outcome from your study, and to incorporate the current literature they mention. Reviewer 3 also makes some excellent recommendations that seem easily accomplished. I know that this review period has been quite long with the difficulty of obtaining reviewers, but I hope this revision is fairly straightforward and we can proceed quickly.

We look forward to receiving your revised manuscript.

Kind regards,

Lorien Shana Jasny

Academic Editor

PLOS ONE

Journal Requirements:

a) Did participants provide their written or verbal informed consent to participate in this study?

4. We note that Figure 1 in your submission contain copyrighted images. All PLOS content is published under the Creative Commons Attribution License (CC BY 4.0), which means that the manuscript, images, and Supporting Information files will be freely available online, and any third party is permitted to access, download, copy, distribute, and use these materials in any way, even commercially, with proper attribution. For more information, see our copyright guidelines: http://journals.plos.org/plosone/s/licenses-and-copyright.

Reviewers' comments:

Reviewer's Responses to Questions

**Comments to the Author**

1. Is the manuscript technically sound, and do the data support the conclusions?

Reviewer #1: Yes

Reviewer #2: Partly

Reviewer #3: Yes

2. Has the statistical analysis been performed appropriately and rigorously? 

Reviewer #1: Yes

Reviewer #2: Yes

Reviewer #3: Yes

3. Have the authors made all data underlying the findings in their manuscript fully available?

Reviewer #1: Yes

Reviewer #2: Yes

Reviewer #3: Yes

4. Is the manuscript presented in an intelligible fashion and written in standard English?

Reviewer #1: Yes

Reviewer #2: Yes

Reviewer #3: Yes

5. Review Comments to the Author

Reviewer #1: Thank you for the opportunity to review this manuscript. The topic of the manuscript is timely, interesting, and fills a noticeable void that exists within the existing literature. The study itself is rigorous, insightful, and leverages a novel research approach to address the authors' research goals. My only minor suggestion would be to synthesize a bit more of the existing literature relative to social connectedness and capital earlier in the paper, as you do not introduce these terms until page 7. All in all, this is a really strong contribution to the existing literature and I am incredibly impressed with the current version of the manuscript.

Reviewer #2: This paper makes an important distinction between (and observation of) the different ways that social media users go about using social media platforms. In particular, I find the validation of self-report measures of usage style with behavioral data and the clustering analysis to be most valuable and interesting. However, I also feel that the paper slightly overstates the possibility that it distinguishes between which of these usage styles is associated with which consequences for well-being, and believe it could benefit from more careful framing of the outcomes of the well-being analysis and of the prior literature in the introduction.

In particular, the discussion of the well-being effects both in the introduction and in parts of the results section lead the reader to presume that the analysis will show that different usage styles leads to different well-being outcomes. However, these well-being “outcomes” are only assessed once–after users engage in a behavioral task. The authors note clearly in the discussion that these styles are only associated with well-being measures–but framing these differences as outcomes of social media use neglects to consider that we don’t know the initial state of users’ well-being (a baseline measure would solve this) before the behavioral task occurred. This also leads to issues with reverse causality, as we don’t know whether, for example, people with higher well being simply have more friends on Facebook, are more active users, and might be predisposed to a particular usage style (in this case, active/engaging).

For example, prior literature (such as Kross et al., 2013) has used ecological momentary assessment (EMA) techniques to evaluate the benefits of social media usage and different behaviors within the social media context. This study might be able to better assess consequences/outcomes of social media use style on well being if it observed users’ behavior more than once and looked at how usage altered users’ feelings of well-being over time, as has been done in much of the prior literature they cite in the introduction. Given this, to be published as the data stands currently, well-being measures in this study should not be considered outcomes or consequences of social media use. The framing of these measures should be consistently addressed in the introduction and in the discussion as well.

A second recommendation in terms of content and study limitations is related to the conceptualization of the study and the use of a novel social media platform/aggregator in the study design rather than observing user behavior in a naturalistic way (e.g. on Facebook itself). A piece of recent literature that may be helpful to the authors, but also may reduce the novelty of these findings, is the emerging literature on social media habits (Bayer et al., 2022; Anderson & Wood, 2021) and reward learning (Lindstrom et al., 2021). Though the authors note that weekend usage hours do impact some well-being measures, we do not have a clear idea of how strong the transfer of users’ Facebook habits (though the authors use the instructions to direct users to behave in the same way they do on other social networks explicitly) to this new platform may impact well-being, if at all. The authors’ analysis neglects the established idea that social platforms are reward-learning machines, and these rewards would be what develop and generate the different styles of social media use which they outline. Using a novel social media platform limits the conclusions of this study to well-being effects of users who are using a brand new social media platform for the very first time, as the platform they have used to observe behavior does not have precisely the same contextual cues that may trigger users’ actual Facebook habits, which (as established in Anderson & Wood, 2021) may be most prominent among users who are frequent social media users (posting, reacting, scrolling) upon perception. I would recommend mentioning this in the discussion of study limitations, and to add a sentence or two to the introduction specifying that the observation procedure may not directly transfer user habits of specific behaviors such as these from the other apps.

Overall, I think that in spite of these limitations, the paper does a number of valuable things, in particular related to helping us understand that usage styles are related to well-being in different ways. This paper extends some of the suggestions from the Bayer et al. (2022) paper cited above, which suggests breaking social media activitiy (scrolling, posting, liking, sharing etc) into component parts to better understand specific well-being outcomes. In addition, the clustering method used is valuable and novel within this research area to my knoweldge, the study provides a nice validation of self-reported social media use styles, as well as suggests that there is a clear relationship between the # of friends a user has on social media and their general well-being, as well as the way they use the site. Finally, the OSF-based materials and data were very well-organized and easily accessible.

Specific issues:

Pp 12– mean of sense of belonging is outside the given range (likely a typo)

Pp 19– could be useful to say that “greater/fewer” usage hours (or other covairate) was significantly associated with increased/decreased sense of belonging (or specifying direction of the association), just to make readability and interpretation easier.

REFERENCES

Anderson, I. A., & Wood, W. (2021). Habits and the electronic herd: The psychology behind social media’s successes and failures. Consumer Psychology Review, 4(1), 83-99. https://doi.org/10.1002/arcp.1063

Bayer, J. B., Anderson, I.A., & Tokunaga, R. (2022). Building and breaking social media habits. Current Opinion in Psychology. https://doi.org/10.1016/j.copsyc.2022.101303

Kross, E., Verduyn, P., Demiralp, E., Park, J., Lee, D. S., Lin, N., ... & Ybarra, O. (2013). Facebook use predicts declines in subjective well-being in young adults. PloS one, 8(8), e69841.

Lindström, B., Bellander, M., Schultner, D. T., Chang, A., Tobler, P. N., & Amodio, D. M. (2021). A computational reward learning account of social media engagement. Nature Communications, 12(1), 1-10. https://doi.org/10.1038/s41467-020-19607-x

Reviewer #3: It’s not what you do, it’s the way that you do it: An experimental task delineates among styles of behaviour on social networking sites and psychosocial measures.

Thanks for giving me the chance to review the manuscript “It’s not what you do, it’s the way that you do it: An experimental task delineates among styles of behaviour on social networking sites and psychosocial measures”. The topic is timely and interesting, and the method and result sections are sound. Paper is well written, and issues raised are relevant. The major strength is that the article is focused on exploring styles of behaviour on social networking sites, especially in an experimental way. This paper proposes that, "different styles of behaviour" is a possible mechanism.

These experimental tasks are relatively new, and their research provides new ideas for researchers in the field of social networking sites behaviour. After data analysis, the author/s discusses the results. The overall research idea is clear and the research process is reasonable. The concept is a potential contribution to our understanding of social networking sites. However, the manuscript needs some revision to make this contribution clearer and stronger.

1. First, when throughout the Introduction section, the authors correctly focused on relevant literature. Paper still needs to include a more theoretical explanation of social networking sites.

2. I agree with their concerns about self-report scales. However, they also used these scales. They need to explain this better.

3. These psychosocial concepts are quite extended in years, what's new in this manuscript?

4. There are some grammatical errors throughout the manuscript that will need to be addressed.

5. The authors stated that they developed mock SNS as a hybrid of Facebook and Instagram. But there is no information about participants' Instagram profiles. Why didn't they explain briefly? I would have to see that either, information on this should also be included.

6. Finally, what is the take-home message of the present manuscript? I would like to see some sort of discussion around the implications of these findings and much more interesting directions for future research.

7. What are, specifically, the psychological variables that need to be better investigated? Here, more details/information are required.

6. PLOS authors have the option to publish the peer review history of their article (what does this mean?). If published, this will include your full peer review and any attached files.

Reviewer #1: **Yes: **Briana Trifiro

Reviewer #2: No

Reviewer #3: No

---

## [Author Response · Author response to Decision Letter 0]

7 Sep 2022

Please see attached file "PONE-D-22-09916 Response to Reviewers" which has better formatting. 

Dear Editor,

We submit a revision of manuscript PONE-D-22-09916, in which we have addressed all of the reviewers’ comments and suggestions. Please note that, in our response to reviewers below, the page and line numbers correspond to the ‘tracked changes’ document (and not the clean document). 

Editor’s Comments

We have now updated our manuscript in line with PLOS ONE’s style requirements.

2. Please amend your current ethics statement to address the following concerns: A) Did participants provide their written or verbal informed consent to participate in this study? B) If consent was verbal, please explain i) why written consent was not obtained, ii) how you documented participant consent, and iii) whether the ethics committees/IRB approved this consent procedure.

This information is provided on page 8, line 9 under the subsection titled “Participants”:

“All volunteers provided written informed consent and the study was approved by the University Research Ethics Committee at Aston University (ref. 1649).”

We have now revised the manuscript accordingly.

4. We note that Figure 1 in your submission contain copyrighted images. All PLOS content is published under the Creative Commons Attribution License (CC BY 4.0), which means that the manuscript, images, and Supporting Information files will be freely available online, and any third party is permitted to access, download, copy, distribute, and use these materials in any way, even commercially, with proper attribution. We require you to either (1) present written permission from the copyright holder to publish these figures specifically under the CC BY 4.0 license, or (2) remove the figures from your submission. 

The Authors of the Nencki Affective Picture System (NAPS) provided written permission, via email correspondence to the first author, for us to use the five example ‘neutral’ images with appropriate citation at the stage of our first submission to PLOS One. They have now completed the Content Permissions Form, which we have uploaded as an “Other” file with our resubmission. We have also updated the figure caption to state: “Reprinted from Marchewka et al. (2014) under a CC BY license, with permission from Artur Marchewka, original copyright 2014”.

For full transparency, please also note that we asked the original Authors of the NAPS whether we could upload the neutral images, which comprised our stimulus images, to the Open Science Framework as ‘open materials’ to which they declined due to copyright. Instead, we provide one of the approved example neutral images within our open materials with a “readme” file documenting what other researchers will need to do in order to reuse the experimental task.

We have reviewed the reference list to ensure it is complete and correct and can confirm that no retracted articles are cited. Changes to the Reference List include only the articles below that were recommended by the reviewers. 

Reviewer #1

1. My only minor suggestion would be to synthesize a bit more of the existing literature relative to social connectedness and capital earlier in the paper, as you do not introduce these terms until page 7. All in all, this is a really strong contribution to the existing literature and I am incredibly impressed with the current version of the manuscript.

The main aim of this study was to introduce and empirically test a novel behavioural task to dissociate interactive, reactive, and passive social networking site use. The secondary aim was to establish whether these dissociable ‘styles’ were associated with psychosocial variables evaluated frequently in the literature. It is for this reason that our manuscript starts with a general overview of the current literature, and its associated measurement issues, before outlining the psychosocial variables. Nevertheless, we agree that this secondary aim is introduced rather late in the Introduction, and so we have revised the initial summary of our aims to include this (page 4, line 10):

“To facilitate the reconciliation of these inconsistencies, the present study developed a computerised task to measure styles of behaviour on a mock SNS – one that can be used in future research to empirically assess the psychosocial outcomes of dissociable styles of usage. To assess the potential utility of the task for this means, we examined whether distinct usage styles differ on several psychosocial variables evaluated commonly as outcome measures in this literature: loneliness, sense of belonging, social connectedness and social capital.”

Reviewer #2

1. [T]he paper slightly overstates the possibility that it distinguishes between which of these usage styles is associated with which consequences for well-being, and believe it could benefit from more careful framing of the outcomes of the well-being analysis and of the prior literature in the introduction. In particular, the discussion of the well-being effects both in the introduction and in parts of the results section lead the reader to presume that the analysis will show that different usage styles leads to different well-being outcomes. However, these well-being “outcomes” are only assessed once–after users engage in a behavioral task. The authors note clearly in the discussion that these styles are only associated with well-being measures–but framing these differences as outcomes of social media use neglects to consider that we don’t know the initial state of users’ well-being (a baseline measure would solve this) before the behavioral task occurred. This also leads to issues with reverse causality, as we don’t know whether, for example, people with higher well being simply have more friends on Facebook, are more active users, and might be predisposed to a particular usage style (in this case, active/engaging). For example, prior literature (such as Kross et al., 2013) has used ecological momentary assessment (EMA) techniques to evaluate the benefits of social media usage and different behaviors within the social media context. This study might be able to better assess consequences/outcomes of social media use style on well being if it observed users’ behavior more than once and looked at how usage altered users’ feelings of well-being over time, as has been done in much of the prior literature they cite in the introduction. Given this, to be published as the data stands currently, well-being measures in this study should not be considered outcomes or consequences of social media use. The framing of these measures should be consistently addressed in the introduction and in the discussion as well.

In response to this comment, and a suggestion from Reviewer #1, we have removed the term ‘outcomes’ throughout when discussing our findings, instead using the term ‘psychosocial variables’ (we only refer to ‘outcomes’ when discussing previous research in this area or recommendations for future research). We have also made the following changes to the Title, Abstract (page 3, line 12), Introduction (page 4, line 10; page 6, line 13), and Discussion (page 22, line 17) to state more clearly the rationale behind our comparisons of psychosocial variables between the styles of usage. In addition, we have revised the title to reflect this more accurately:

“It’s not what you do, it’s the way that you do it: An experimental task delineates among passive, reactive and interactive styles of behaviour on social networking sites.”

“Furthermore, our data reveal that these usage styles differ on several measures of psychosocial variables employed frequently in the disparate literature: more interactive users reported greater feelings of social connectedness and social capital than passive or reactive users.”

“To facilitate the reconciliation of these inconsistencies, the present study developed a computerised task to measure styles of behaviour on a mock SNS – one that can be used in future research to empirically assess the psychosocial outcomes of dissociable styles of usage. To assess the potential utility of the task for this means, we examined whether distinct usage styles differ on several psychosocial variables evaluated commonly as outcome measures in this literature: loneliness, sense of belonging, social connectedness and social capital.”

“To investigate if these more nuanced styles of usage can be identified behaviourally, the present study developed the Social Networking Site Behaviour Task (SNSBT) – a mock social networking platform based loosely on Facebook and Instagram. As a means of validating the task, we then examined if dissociable patterns of reactive and interactive behaviour converged with subjective perceptions of usage style assessed with a self-report instrument based on the active-passive dichotomy (Escobar-Viera et al., 2018), which we modified to further differentiate between reactive and interactive usage. Finally, we examined whether distinct usage styles identified on this mock SNS differed in terms of several psychosocial factors employed frequently in the disparate literature. Importantly, this was not to assess the outcomes of dissociable styles, which requires pre- and post-usage comparisons or real-time measurements (e.g., (Kross et al., 2013); rather, this evaluated the potential utility of the SNSBT for future research and reconciling inconsistent findings. First we assessed differences in self-reported loneliness and sense of belonging…”

“Individuals exhibiting more interactive usage reported stronger feelings of connectedness and social capital compared with those who showed more reactive or passive behaviour, and those with a more reactive style of usage reported feeling more connected than those expressing a tendency for passive usage. It is important to stress that these results of our comparisons between usage styles on subjective ratings of psychosocial variables provides only a measure of the potential utility of this novel task for future research; they provide no indication of the outcomes of SNSBT usage styles. To measure the outcomes of different usage styles on this task, future studies should employ the experimental designs used elsewhere; for example, they might measure changes in users’ mood (Mosquera et al., 2020; Sagioglou & Greitemeyer, 2014) or subjective well-being (Allcott et al., 2020; Verduyn et al., 2015) before and after specific types of engagement on the task. Alternatively, to capture the outcomes of spontaneous usage styles, future research might employ real-time ecological measurements of psychosocial variables during engagement with the SNSBT (see Kross et al., 2013).”

2. A second recommendation in terms of content and study limitations is related to the conceptualization of the study and the use of a novel social media platform/aggregator in the study design rather than observing user behavior in a naturalistic way (e.g. on Facebook itself). A piece of recent literature that may be helpful to the authors, but also may reduce the novelty of these findings, is the emerging literature on social media habits (Bayer et al., 2022; Anderson & Wood, 2021) and reward learning (Lindstrom et al., 2021). Though the authors note that weekend usage hours do impact some well-being measures, we do not have a clear idea of how strong the transfer of users’ Facebook habits (though the authors use the instructions to direct users to behave in the same way they do on other social networks explicitly) to this new platform may impact well-being, if at all. The authors’ analysis neglects the established idea that social platforms are reward-learning machines, and these rewards would be what develop and generate the different styles of social media use which they outline. Using a novel social media platform limits the conclusions of this study to well-being effects of users who are using a brand new social media platform for the very first time, as the platform they have used to observe behavior does not have precisely the same contextual cues that may trigger users’ actual Facebook habits, which (as established in Anderson & Wood, 2021) may be most prominent among users who are frequent social media users (posting, reacting, scrolling) upon perception. I would recommend mentioning this in the discussion of study limitations, and to add a sentence or two to the introduction specifying that the observation procedure may not directly transfer user habits of specific behaviors such as these from the other apps.

We thank the reviewer for bringing this important literature to our attention. As suggested, we have incorporated into the revised Discussion an acknowledgement that many of the cues and contextual factors believed to reinforce and encourage social media habits are currently lacking from the initial version of the SNSBT (page 25, line 21): 

“Finally, although the current instantiation of the SNSBT incorporated some characteristics of real SNS platforms (i.e., content scrolling, liking, sharing), it did not expose users to other important design features that have been shown to cue habitual SNS usage patterns. For example, the hybrid layout of the mock platform differed from either Facebook or Instagram, thereby removing contextual cues that might activate stimulus-response associations that drive particular usage styles (e.g., within-platform chat sounds). Similarly, the non-emotional landscape images used in the current version of the task differ considerably from the unexpected and often emotional content through which users scroll on real SNS platforms, which are considered to be more rewarding and capable of triggering habitual behaviours acquired through reinforcement learning (Anderson & Wood, 2021; Bayer et al., 2022; Lindström et al., 2021). As such, the dissociable styles of usage that we have identified with the SNSBT may not perfectly reflect users’ real-world habits on SNS platforms. However, the ease with which the layout, interface and content of the SNSBT can be modified presents a tool with which these potential drivers of social media habit formation can be investigated experimentally.”

3. Pp 12– mean of sense of belonging is outside the given range (likely a typo).

We thank the reviewer for bringing this oversight to our attention. It is in fact the range that is incorrect and not the mean; specifically, the scale is anchored between 1-4 with 18-items, meaning the lowest range can be 18 and the highest 72. The mean across the present sample was 54.06 (SD = 13.21) but the range was 19–72 (we had mistakenly reported “27” as the upper range). This has now been corrected in the manuscript.

4. Pp 19– could be useful to say that “greater/fewer” usage hours (or other covariate) was significantly associated with increased/decreased sense of belonging (or specifying direction of the association), just to make readability and interpretation easier.

The specific directions of difference between clusters (groups) in both Friends and Hours is stated explicitly when reporting the results of the ANOVA and non-parametric analyses of these two dependent variables (page 18, line 3):

“Individuals comprising the Interactive cluster reported significantly more Friends (486.65 [526.85]) than those the Passive cluster (341.58 [345.43], pcorr=.010, d=.33), while members of the Reactive cluster (444.98 [477.28]) did not differ significantly from their Interactive (pcorr>.999, d=.08) or Passive counterparts (pcorr=.067, d=.25). Furthermore, a non-parametric Kruskall-Wallis test demonstrated that the clusters differed significantly in the ordinal number of weekend hours they spent on Facebook (H[2]=35.01, p<.001, ε2=.06]). Follow-up Mann-Whitney tests showed that the Interactive cluster reported spending more Hours on this SNS at the weekend (median=2-3 hours) than the Reactive (median=1-2 hours; Z=2.56, p<.001) and Passive clusters (median=1-2 hours; Z=5.69, p<.001), and the Reactive cluster reported more Hours than the Passive users (Z=3.70, p<.001). ”

Since this paragraph specifies the direction of difference (Friends = Interactive>[Reactive & Passive]; Hours = Interactive>Reactive>Passive), we have not repeated this information when reporting the results from the exploratory ANCOVA.

Reviewer #3:

1. [W]hen throughout the Introduction section, the authors correctly focused on relevant literature. Paper still needs to include a more theoretical explanation of social networking sites.

Providing a comprehensive review of theoretical explanations for social networking sites (SNS) is beyond the scope of this paper, which provides an empirical investigation of social networking site usage based upon one such theoretical explanation - the interpersonal-connection-behaviours framework (Clark et al., 2018). Instead, throughout the paper we refer readers to several reviews for more comprehensive theoretical discussions of SNS behaviours and outcomes. Nevertheless, this and other comments have helped us to clarify several aspects of the paper: we now provide common examples of SNS platforms in the opening sentence of the revised Introduction, and in the revised Discussion we outline an emerging theory of habitual SNS usage and reward contingency (e.g., Anderson & Wood, 2002; Bayer et al, 2022). 

2. I agree with their concerns about self-report scales. However, they also used these scales. They need to explain this better.

We thank the reviewer for making us aware of this apparent contradiction in the original manuscript. To avoid this, we have revised the Introduction to make clearer our intended use of the (modified) questionnaire of SNS usage; namely, to assess the degree of convergence in our novel behaviour measure and subjective perceptions of passive, reactive and interactive styles (page 5, line 9; page 6, line 13).

“A recent meta-analysis has revealed the vast number of self-report instruments that have been developed for measuring styles of SNS usage along the active-passive dichotomy (Valkenberg et al., 2022). Recently, however, many scholars have questioned the usefulness of this crude categorization, calling for a more nuanced refinement (e.g., Kross et al., 2021). Furthermore, it is argued that the field of cyberpsychology needs to move beyond subjective self-report questionnaires and consider behavioural methods that allow for more direct assessments of usage styles on SNS (Parry et al., 2021).”

“To investigate if these more nuanced styles of usage can be identified behaviourally, the present study developed the Social Networking Site Behaviour Task (SNSBT) – a mock social networking platform based loosely on Facebook and Instagram. As a means of validating this task, we then examined if dissociable patterns of reactive and interactive behaviour converged with subjective perceptions of usage style assessed with a commonly used self-report instrument based on the active-passive dichotomy (Escobar-Viera et al., 2018), which we modified to further differentiate between reactive and interactive usage.”

3. These psychosocial concepts are quite extended in years, what's new in this manuscript?

It is important to stress that we incorporated an assessment of these psychosocial variables simply to evaluate the potential utility of our novel task for future research, not to examine the psychosocial outcomes from usage of this task. To make this contribution to the literature much clearer, we have clarified our rationale for using these psychosocial measures (see our response to comment 1 from reviewer #2).

We agree that these psychosocial variables have been investigated frequently in the (cyber)psychology literature; indeed, it is for this reason that we selected these particular variables. However, studies have produced vastly disparate findings when exploring the relationship between SNS usage and psychosocial well-being. Our novel task therefore sheds new light on how these inconsistencies might reflect a lack of consideration for how people use SNS, and the utility of our novel behavioural task for future research into the outcomes of dissociable usage styles. 

4. There are some grammatical errors throughout the manuscript that will need to be addressed.

All co-authors have proof-read the revised manuscript thoroughly to correct any grammatical errors.

5. The authors stated that they developed mock SNS as a hybrid of Facebook and Instagram. But there is no information about participants' Instagram profiles. Why didn't they explain briefly? I would have to see that either, information on this should also be included.

Facebook is the most popular SNS platform worldwide, and far more so than either Instagram or Twitter (Perrin & Anderson, 2019; Pew Research Center). It is for this reason that we used Facebook engagement as an inclusion criterion for this study, and Facebook parameters (e.g., number of Facebook Friends, hours spent on Facebook in an average weekday or weekend) as self-report metrics of usage. Due to copyright restrictions, we were unable to incorporate certain design features of Facebook into the novel computerised task; this included the Facebook “like” symbol. To overcome this, we created a new layout and custom symbols that would still be somewhat familiar to participants (e.g., a red heart resembling the “like” symbol of Instagram). This is why we describe the layout of the computerised as being “based loosely on Facebook and Instagram”. For participants, however, the layout would appear as a new SNS platform, as outlined in the Methods Instructions.

6. Finally, what is the take-home message of the present manuscript? I would like to see some sort of discussion around the implications of these findings and much more interesting directions for future research.

We feel that by addressing each of the reviewers’ comments above, the main take-home message of this study has been made much clearer - specifically, this research demonstrates the potential utility of a new computerised task capable of dissociating among styles of SNS usage. In response to this comment, we have made the following revisions to the Conclusion (page 25, line 6):

“We have developed a simple, publicly available computerised task that provides a new method with which to objectively quantify different styles of behaviour on social networking sites (SNS). The versatility of this task enhances its utility for future research in the field of cyberpsychology; it can be adapted easily to measure usage styles across multiple SNS platforms, to assess content and design features of SNS that might influence discrete usage styles, and perform experimental investigations into the influence of different SNS usage styles on psychosocial outcomes. ”

7. What are, specifically, the psychological variables that need to be better investigated? Here, more details/information are required.

In light of the primary goal of this study - that is, to develop and evaluate the utility of a new experimental method for cyberpsychology research that provides behavioural measures of SNS usage - we offer only tentative and speculative suggestions on the psychological mechanisms behind different usage styles different (page 22, line 17; page 23, line 4):

“Individuals exhibiting more interactive usage reported stronger feelings of connectedness and social capital compared with those who showed more reactive or passive behaviour, and those with a more reactive style of usage reported feeling more connected than those expressing a tendency for passive usage. It is important to stress that these results of our comparisons between usage styles on subjective ratings of psychosocial variables provides only a measure of the potential utility of this novel task for future research; they provide no indication of the outcomes of SNSBT usage styles. To measure the outcomes of different usage styles on this task, future studies should employ the experimental designs used elsewhere; for example, they might measure changes in users’ mood (Mosquera et al., 2020; Sagioglou & Greitemeyer, 2014) or subjective well-being (Allcott et al., 2020; Verduyn et al., 2015) before and after specific types of engagement on the task. Alternatively, to capture the outcomes of spontaneous usage styles, future research might employ real-time ecological measurements of psychosocial variables during engagement with the SNSBT (see (Kross et al., 2013).”

“[O]ur results also revealed that while psychosocial variables differ among individuals according to their response tendencies on the SNSBT, this was often accounted for by the number of friends that participants reported having on Facebook and the amount of time they spent on the platform. [This might] indicate that usage style operates in concert with, or even emerges as a function of, online network size and time spent on SNS. Indeed, psychological theories of friendship (e.g., Dunbar, 2018; Oswald et al., 2004) posit that while social networks provide an opportunity for emotional closeness and social support, the degree to which an individual experiences this is dependent upon the intensity and intimacy of relationships they maintain with their network. Larger online social networks reported by interactive users will enable them to have more meaningful (bidirectional) exchanges with a wider network of close and casual friends, allowing them to exercise their preferred style more frequently and benefit from it maximally.”

Finally, please note that the order of authorship has changed slightly to accurately reflect each Author’s contribution to this revised article; specifically, Dr Charlotte R. Pennington is now the last author. We have updated this accordingly on the submission portal.

We would like to take this time to thank the reviewers for their positive and insightful feedback, which we believe has improved the manuscript considerably. We hope you share our enthusiasm and look forward to hearing from you in due course.

Sincerely,

Dr Daniel Shaw and co-authors.

---

## [Editor Report · Decision Letter 1]

13 Oct 2022

It’s not what you do, it’s the way that you do it: An experimental task delineates among passive, reactive and interactive styles of behaviour on social networking sites.

PONE-D-22-09916R1

Dear Dr. Shaw,

We’re pleased to inform you that your manuscript has been judged scientifically suitable for publication and will be formally accepted for publication once it meets all outstanding technical requirements.

Kind regards,

Lorien Shana Jasny

Academic Editor

PLOS ONE
---

## [Editor Report · Acceptance letter]

2 Nov 2022

PONE-D-22-09916R1 

It’s not *what* you do, it’s the *way* that you do it: An experimental task delineates among passive, reactive and interactive styles of behaviour on social networking sites. 

Dear Dr. Shaw:

I'm pleased to inform you that your manuscript has been deemed suitable for publication in PLOS ONE. Congratulations! Your manuscript is now with our production department. 

Kind regards, 

on behalf of

Dr. Lorien Shana Jasny 

Academic Editor

PLOS ONE